# Efficacy of Granulocyte Colony-Stimulating Factor in Acute on Chronic Liver Failure: A Systematic Review and Survival Meta-Analysis

**DOI:** 10.3390/jcm12206541

**Published:** 2023-10-16

**Authors:** Georgios Konstantis, Georgia Tsaousi, Chryssa Pourzitaki, Elisavet Kitsikidou, Dimitrios E. Magouliotis, Sebastian Wiener, Amos Cornelius Zeller, Katharina Willuweit, Hartmut H. Schmidt, Jassin Rashidi-Alavijeh

**Affiliations:** 1Clinical Pharmacology, Faculty of Medicine, School of Health Sciences, Aristotle University of Thessaloniki, 54124 Thessaloniki, Greece; chpour@gmail.com; 2Department of Gastroenterology, Hepatology and Transplant Medicine, Medical Faculty, University of Duisburg-Essen, 40219 Essen, Germany; 3Department of Anesthesiology and ICU, Medical School, Aristotle University of Thessaloniki, 54124 Thessaloniki, Greece; tsaousig@otenet.gr; 4Department of Internal Medicine, Evangelical Hospital Dusseldorf, 40217 Dusseldorf, Germany; eliskits@gmail.com; 5Department of Surgery, University of Thessaly, Biopolis, 41110 Larissa, Greece

**Keywords:** liver failure, failure, acute on chronic liver (ACLF), acute-on-chronic liver failure, granulocyte colony-stimulating factor (G-CSF)

## Abstract

Background: Acute-on-chronic liver failure (ACLF) mostly occurs when there is an acute insult to the liver in patients with pre-existing liver disease, and it is characterized by a high mortality rate. Various therapeutic approaches have been used thus far, with orthotopic liver transplantation being the only definitive cure. Clinical trials and meta-analyses have investigated the use of granulocyte colony-stimulating factor (G-CSF) to mobilize bone marrow-derived stem cells. Some studies have suggested that G-CSF may have a significant role in the management and survival of patients with ACLF. However, the results are conflicting, and the efficacy of G-CSF still needs to be confirmed. Aim: The aim was to assess the efficacy of G-CSF in patients with ACLF. Methods: Electronic databases were searched until May 2023 for randomized controlled trials investigating the use of G-CSF in adult patients with ACLF. Outcome measures were the effects of G-CSF on overall survival, changes in liver disease severity scores, complications of cirrhosis, other G-CSF-related adverse effects, and all-cause mortality. The study’s protocol has been registered with Prospero (CRD42023420273). Results: Five double-blind randomized controlled trials involving a total of 421 participants met the inclusion criteria. The use of G-CSF demonstrated a significant effect on overall survival (HR 0.63, 95% CI 0.41 to 0.95, and I^2^ 48%), leading to a decreased mortality (LogOR-0.97, 95% CI −1.57 to −0.37, and I^2^ 37.6%) and improved Model for End-Stage Liver Disease (MELD) scores (SMD −0.87, 95% CI −1.62 to −0.13, and I^2^ 87.3%). There was no correlation between the improvement of the Child–Pugh score and the use of G-CSF(SMD −2.47, 95% CI −5.78 to 0.83, and I^2^ 98.1%). The incidence of complications of cirrhosis did not decrease significantly with G-CSF treatment (rate ratio 0.51, 95% CI 0.26 to 1.01, and I^2^ 90%). A qualitative synthesis showed that the use of G-CSF is safe. Conclusions: The administration of G-CSF has demonstrated a positive impact on overall survival, liver function, and the MELD score. The presence of heterogeneity in the included studies prohibits conclusive recommendations.

## 1. Introduction

Acute-on-chronic liver failure (ACLF) is a term under which extensive pathologies and symptom constellations can be summarized [1]. It was recently recognized as a single entity, and since then, numerous reports and clinical studies in the literature have attempted to establish precise terminology and diagnostic criteria [2]. The most commonly used diagnostic criteria are the European Association for the Study of the Liver (EASL)-CLIF and the Asian Pacific Association for the Study of the Liver (APASL) criteria [2]. Both focus on the presence of an underlying chronic hepatopathy that is decompensated by a triggering event and results in the failure of one or more organs. However, there are significant differences between the two. The EASL-CLIF criteria define ACLF [1] as an acute impairment of liver function in the setting of underlying liver cirrhosis, whereas the APASL criteria [3] only necessitate the presence of a chronic liver disease. 

The natural course of the disease is complicated by the clinical and biochemical features of systemic inflammatory response syndrome (SIRS) [4]. Elevated levels of various pro- and anti-inflammatory cytokines, including TNF-α, IL-2, IL-6, IL-8, IL-10, and IFN-γ, have been identified [5,6], suggesting a pronounced immune response that leads to immune paralysis, which further increases the risk of infection and inflammation. This immune pathology in ACLF resembles the multimodal immune response seen in patients with severe sepsis, starting with an initial SIRS response, followed by a mixed and then a compensated anti-inflammatory response (MARS and CARS, respectively) [7]. In this context, various therapies targeting immune paralysis have been investigated. Among these approaches, bone-marrow-derived stem cells (BMSCs) have been shown to modulate immune functions in patients with ACLF and, in addition, promote liver tissue regeneration [8]. 

Liver regeneration is a complex process involving the regeneration of mature liver cells, the activation and proliferation of hepatic progenitor cells, and the recruitment of bone-marrow-derived stem cells [9]. These pluripotent cells have the ability to transform into various cell types and can contribute significantly to liver regeneration, leading to accelerated healing [10,11]. Several studies, both clinical and preclinical, have demonstrated that the administration of these cells, either systemically or directly into the liver artery, can improve tissue repair [11,12,13]. However, their administration is associated with certain difficulties and complications, making them less than an ideal clinical solution. Granulocyte colony-stimulating factor (G-CSF) is a glycoprotein that plays a crucial role in stimulating the production, maturation, and mobilization of neutrophils [14], and it has the ability to recruit bone marrow cells by binding to specific receptors expressed on many liver regeneration and bone marrow cells, leading to similar effects [15,16]. Experimental studies have shown that bone marrow cells recruited by G-CSF can be found in liver cells, suggesting that the administration of G-CSF could have a beneficial effect on liver regeneration [16,17,18]. The examination and confirmation of the safety and clinical benefits of G-CSF in ACLF patients could offer another valuable tool in the fight against ACLF. It has the potential to serve as a “bridging to recovery” or even “bridging to transplantation” strategy, significantly enhancing the chances of survival for ACLF patients, within the realm of non-invasive treatment options.

Nonetheless, clinical studies are inconclusive, and a recent European study [19] did not show any benefit of the G-CSF administration compared with standard medical treatment. Based on the findings of this study, the recently published European guidelines [20] advise against the administration of G-CSF to patients with ACLF. Previous meta-analyses included patients with acute alcoholic hepatitis [21] and decompensated cirrhosis [22], which may have introduced bias into the data synthesis. The aim of this systematic review and meta-analysis is to comprehensively assess the available clinical data on the potential benefits of G-CSF therapy in patients with ACLF. Specifically, we evaluated the impact of G-CSF therapy on overall survival, mortality, cirrhosis- and G-CSF-related complications, and improvements in MELD and Child–Pugh scores. To our knowledge, this is the first state-of-the-art systematic review and meta-analysis that focuses solely on the effect of G-CSF on patients specifically with ACLF, as described in the APASL and EASL guidelines. The use of HR (hazard ratio) as an effective measure was chosen over OR (odds ratio) for survival analysis due to its suitability for capturing survival-related outcomes.

## 2. Materials and Methods

This systematic review and meta-analysis were conducted in accordance with the PRISMA statement [23]. To identify relevant studies, a systematic and comprehensive search was conducted in electronic databases MEDLINE, SCOPUS, and Cochrane database from February 2023 to May 2023. To increase the potential search results, no language restrictions were applied. The search strategy is thoroughly presented in Appendix A. ACLF was defined in the randomized controlled trial (RCT) selection process using the EASL-CLIF and APASL criteria [1,3]. The study protocol was registered at Prospero (registration number CRD42023420273).

### 2.1. Eligibility Criteria

This meta-analysis included randomized controlled trials (RCTs) that compared the efficacy of G-CSF versus placebo or any other intervention in adult patients with ACLF, and data were reported for the primary outcome, i.e., overall survival. Studies that investigated the use of G-CSF in combination with other active medications were excluded to focus specifically on the effects of G-CSF. Studies that enrolled adults with acute liver failure not meeting the criteria for ACLF or patients with acute alcoholic steatohepatitis were also excluded. The APASL and EASL-CLIF definitions differ in their emphasis on liver failure and extra-hepatic organ failure, respectively. The definition of ACLF by EASL-CLIF is based on the CANONIC study [24] and includes three major characteristics: acute decompensation, organ failure, and a high 28-day mortality rate. On the other hand, the APASL criteria define ACLF-CLIF as an acute hepatic insult leading to jaundice and coagulopathy, complicated within four weeks by ascites and/or encephalopathy, in patients with chronic liver disease or cirrhosis, and high 28-day mortality. ACLF was defined using either the ACLF-EASL or APASL criteria.

### 2.2. Data Collection and Extraction

Suitable records were imported into Endnote 19(Clarivate, London, United Kingdom), and duplicates were removed. Two independent reviewers (GK and EK) screened the records at the title and abstract level and then examined eligible studies in full text. Any disagreement during study selection was resolved by a third reviewer (CP). Data regarding publication year, follow-up period, G-CSF dose and administration scheme, baseline characteristics, participant number, standard medical treatment (SMT), and data concerning the primary and secondary endpoints were extracted into a pre-specified form.

### 2.3. Quality Assessment

The risk of bias for the primary outcomes was assessed using the Cochrane Risk of Bias (ROB) tool 2.0 [25]. Two independent reviewers evaluated the included studies (GK and EK), and any disagreement was resolved by a third reviewer (CP). The ROB 2.0 tool evaluates the risk of bias based on factors such as randomization, deviations from intended interventions, missing outcome data, reporting of results, and outcome measurement. If a study met all domains as low risk, it was categorized as low risk, and if it met any domain as high risk, it was considered high risk. In all other cases, the risk of bias was appraised as having some concerns. 

### 2.4. Outcome Measurements

The primary outcome was considered as the effect of G-CSF therapy on overall survival in patients with ACLF for the maximum available follow-up period. Secondary outcomes were considered as the change in liver disease severity scores (Model for End-Stage Liver Disease (MELD) and Child–Pugh (CTP)), complications of cirrhosis, other G-CSF-related adverse effects, and all-cause mortality. Data on G-CSF dosage, the severity of cirrhosis, and other details were also collected. 

### 2.5. Statistical Analysis

To estimate overall survival, the hazard ratio (HR) with a 95% confidence interval (CI) was used. Data were extracted from the studies using the methods described by Tierley et al. [26]. The primary meta-analysis of overall survival included the data from the maximum follow-up period reported in each included study. For continuous data, standardized mean difference (SMD) with 95% CIs was calculated using Hedges’ d estimation method [27]. An analysis of final values was performed because all included studies were randomized clinical trials, and there was no major difference regarding the mean MELD and CTP scores between studies. The main analyses of continuous data were based on the assumption that the last observation was carried forward. To prove the reliability of our data, we performed post hoc sensitivity analysis, including only analysing the available cases in which participants were excluded from primary analysis due to protocol indiscipline, lost to follow-up, or other reasons; these were assumed to be missing at random and were ignored. Complications of cirrhosis were analysed as count-and-rate data by calculating person years of follow-up from each study’s data and graph. Rate ratios (RRs) with 95% CIs were then expressed using the methods outlined by Cochrane [28]. For dichotomous data, odds ratios (OR) with 95% CIs were calculated using the restricted maximum likelihood method. In greater detail, the effect measures selected for our analysis are as follows: The hazard ratio is utilized in survival analysis for comparing survival times between two different groups of patients. The logarithm of the odds ratio is employed to gauge associations between groups when the event under examination is dichotomous. A positive LogOR indicates a higher risk in the first group than in the second group, while a negative LogOR suggests a lower risk. Meanwhile, the standardized mean difference helps measure differences between groups in studies when dealing with continuous outcomes, taking into account standard deviations. Finally, the rate ratio plays a crucial role in epidemiological studies, serving as a metric for comparing event rates between two distinct groups.

Cochrane’s Q was used to explore between-study heterogeneity, and the I^2^ statistic was used to quantify heterogeneity, with a cut-off of 60% or more indicating high heterogeneity. A random effect formula was used to account for differences in methodology and participant characteristics between studies. To explore clinical heterogeneity, sensitivity analyses were performed, taking into account variables such as risk of bias, country of origin, and criteria used to define ACLF. The assessment of publication bias was planned in case more than 10 studies would be retrieved. The statistical analyses were conducted using STATA SE (version 16.1, USA, StataCorp) and Review Manager (RevMan 5.3, Nordic Cochrane Center, Copenhagen, Denmark). 

## 3. Results

### 3.1. Search Results

Five randomized clinical studies were included in our systematic review [19,29,30,31,32]. Table 1 provides an overview of the basic features of the studies and their participants, while Figure 1 depicts the study’s selection procedure and the reasons for exclusion.

### 3.2. Baseline Characteristics

Five studies with a total of 421 patients with ACLF were included. ACLF was defined according to the APASL criteria in four studies and according to the EASL-CLIF criteria in one study [19]. The main differences between the two criteria are presented in Appendix A. Of note, the study that applied the EASL-CLIF criteria was the only one conducted in Europe [19]. The follow-up period varied from 60 to 360 days, while G-CSF was administered for 6 to 26 days at different intervals. The dose of G-CSF was 5 μg/kg, and G-CSF was compared with standard medical treatment (SMT) in all studies. SMT differed between studies and included, among other things, the use of diuretics, lactulose for hepatic encephalopathy, albumin administration, antibiotics for symptoms of infection, and antiviral drugs when the underlying liver disease was of viral etiology (Table 2).

### 3.3. Risk of Bias in the Included Studies

The five studies were assessed for risk of bias by two independent reviewers using the ROB 2.0 for overall survival and changes in liver disease severity scores (MELD and Child–Pugh). The results are presented in Appendix A. A total of three studies were considered low-RoB studies [19,29,32], while two studies were characterized as having some concerns [30,31]. Publication bias due to an inadequate number of studies could not be estimated.

### 3.4. Analysis of Primary Outcomes

#### 3.4.1. Overall Survival

The impact of G-CSF on overall survival was evaluated in all five studies, including 421 patients. One study followed the patients’ overall survival for 360 days [19], while all other studies provided data for shorter time periods ranging from 60 to 180 days. Most studies presented survival data in the form of Kaplan–Meier plots, and data extraction was conducted following the methods reported by Tierley et al. [26]. Patients with ACLF who received G-CSF therapy had a statistically significant survival benefit compared to patients in the SMT group (HR 0.63, CI 95% 0.41 to 0.95, and I^2^ 48%) Figure 2. 

#### 3.4.2. Sensitivity Analyses

In the sensitivity analysis, which only included studies with a low risk of bias [19,29,32], no significant effect of G-CSF could be detected (HR 0.69, CI 95% 0.41 to 1.16, and I^2^ 65%) (Appendix A). Excluding the study from Engelmann et al. [19] resulted in a greater improvement in OS (HR 0.50, CI 95% 0.34 to 0.72, I^2^ 0%) (Appendix A). Taking into account the great heterogeneity of the follow-up period, we proceeded with a subgroup analysis, reporting outcome data at 30, 60, and 90 days. The administration of G-CSF did not result in an improvement in overall survival at 30 and 90 days (HR 0.47, 95% CI 0.19 to 1.20, I^2^ 80% and HR 0.67, 95% CI 0.42 to 1.07, I^2^ 50%), as shown in Appendix A. However, interestingly, a relevant improvement in survival was observed after 60 days (HR 0.61, 95% CI 0.39 to 0.94, I^2^ 50%) (Appendix A). Furthermore, using the data from Engelmann et al. [19], in which competing risk analyses between liver transplantation and death were conducted, the main findings were confirmed (HR 0.63, 95% CI 0.40 to 0.99, I^2^ 80% and HR 0.68, 95% CI 0.43 to 1.07, I^2^ 57%) (Appendix A). 

### 3.5. Analysis of Secondary Outcomes

#### 3.5.1. Change in Liver Disease Severity Scores

##### MELD Score

One study reported the percentage changes in the median MELD value [30]: One study reported changes in the form of a diagram [19], while two studies reported the final values [29,31]. G-CSF showed a beneficial effect compared to all other interventions (SMD −0.87, CI 95% −1.62 to −0.13, I^2^ 87.3%) (Figure 3).

##### Sensitivity Analyses

Considering that two studies reported MELD changes at various time intervals, the main analysis was performed using data from the maximum observation period. A sensitivity analysis using 30-day data was also performed. The results suggested a less pronounced effect (SMD −0.65, CI 95% −1.40 to 0.09, I^2^ 87.8%) (Appendix A). However, the results could not be validated when studies with low RoB were analyzed. (SMD −0.49, CI 95% −1.45 to 0.46, I^2^ 89.3%) (Appendix A). In an attempt to explain the between-study heterogeneity, the only study using the EASL-CLIF criteria was excluded [19]. As a result, the between-study heterogeneity showed a significant improvement, and the results could also be confirmed (SMD −1.20, CI 95% −1.80 to −0.59, I^2^ 62.4%) (Appendix A).

##### Post hoc Analysis

Saha et al. [31] reported MELD scores at 90 days after recruitment only for survivors. A post hoc sensitivity analysis taking this into account was performed, and the results were in accordance with the main analysis (SMD −0.87, CI 95% −1.63 to −0.11, I^2^ 86.7%) (Appendix A).

##### Child–Pugh Score

Only three studies [29,30,31] were included in the evaluation of the effect of G-CSF on CTP. G-CSF showed a trend towards improvement in CTP without achieving statistical significance (SMD −2.47, CI 95% −5.78 to 0.83, I^2^ 98.1%) Figure 4.

##### Sensitivity Analyses

Due to the absence of low-bias studies, it was not possible to perform a sensitivity analysis considering RoB2 assessment. To investigate the heterogeneity between studies, we carried out a re-analysis, where the single study [30] that displayed data as percentage median changes was excluded. The analysis suggested a significant effect, and heterogeneity was significantly improved (SMD −0.76, CI 95% −1.19 to −0.33, I^2^ 0.0%) (Appendix A). When we included data that referred to a 30-day follow-up period, the analysis did not show any differences from the primary analysis (SMD −1.41, CI 95% −3.51 to 0.70, I^2^ 96.4%) (Appendix A).

##### Post hoc

In a post hoc sensitivity analysis, using the available case CTP score from the study of Saha et al. [31], the results of our primary analysis were not altered (SMD −2.49, CI 95% −5.79 to 0.81, I^2^ 97.7%) (Appendix A).

### 3.6. Mortality

All-cause mortality was also evaluated as binary data in all five studies. When the longest observation period of the respective studies was used, the administration of G-CSF was associated with improved survival (LogOR −0.97, CI 95% −1.57 to −0.37, I^2^ 37.6%) (Figure 5).

### 3.7. Sensitivity Analyses

Results were substantiated when the study using the EASL-CLIF criteria [19] was excluded (logOR −1.19, CI 95% −1.75 to −0.63, I^2^ 0) (Appendix A). Results were also supported by excluding studies [30,31] with a high risk of bias (logOR −0.64, CI 95% −1.17 to −0.11, I^2^ 8.3%) (Appendix A).

### 3.8. Complications of Cirrhosis

All included studies evaluated the efficacy of SMT plus G-CSF based on the incidence of serious complications of cirrhosis, such as the development of ascites, variceal rupture, hepatic encephalopathy, and serious infections. According to the meta-analysis, the use of G-CSF in patients with ACLF showed a rate ratio of 0.51 (95% CI 0.26–1.01, I^2^ 90%) for complications of cirrhosis (Figure 6). 

### 3.9. Sensitivity Analysis

Taking into account the risk of bias, the sensitivity analysis demonstrated a relative risk of 0.84 (95% CI 0.55–1.28, I^2^ 75%) (Appendix A). When excluding the study by Engelmann et al. [19], there was a statistically significant reduction in complications of cirrhosis with a rate ratio of 0.40 (95% CI 0.19–0.83, I^2^ 79%) (Appendix A).

### 3.10. G-CSF-Related Adverse Effects

In general, the administration of G-CSF therapy was well tolerated, although limited data prevented statistical analysis. The study conducted by Garg et al. [30] revealed that two patients experienced a transient rash and high fever, leading to the omission of one dose. Another patient in the same study had a reactivation of herpes zoster, which was effectively treated with acyclovir by the end of the study period. Minor side effects, such as fever, headache, and nausea, were reported by Duan et al. [29] while Tong et al. [32] documented a single case of mild rash that resulted in the discontinuation of G-CSF treatment for that specific patient. Importantly, Engelmann et al. [19] registered a considerable number of G-CSF-related adverse events. Out of the reported adverse events, seven were categorized as serious and led to the death of three patients. The causes of death were multiple organ dysfunction in combination with spontaneous bacterial peritonitis, respiratory failure, and acute kidney failure. It was suggested that these conditions were possibly aggravated by the study treatment.

## 4. Discussion

ACLF, regardless of the criteria used for diagnosis, is characterized by a high mortality rate, and currently, liver transplantation appears to be the only definitive therapeutic option [24]. Several clinical studies [29,30,31,32] have shown encouraging results regarding the use of G-CSF in these patients. However, these results are only evident in studies conducted in Asian countries. The only large clinical study conducted in a European setting [19] did not confirm this therapeutic benefit. The aim of this systematic review and meta-analysis was to assess the efficacy of G-CSF administration only in patients with ACLF and to examine the effect of G-CSF in a meta-analysis of survival data. Compared to SMT, G-CSF administration led to improved overall survival and MELD score and reduced mortality. G-CSF was not found to be more effective than SMT in improving the Child–Pugh score and only demonstrated a trend towards reducing the incidence of complications of cirrhosis. 

This systematic review and meta-analysis represent the most recent and comprehensive analysis of the efficacy and safety of G-CSF treatment in patients with ACLF. All studies included in the analysis were conducted with a randomized design, ensuring the reliability of the results and their usefulness in clinical practice. Three additional meta-analyses [21,33,34] have addressed this topic. Nonetheless, these studies may have limitations, such as heterogeneity in patient populations and treatment protocols, and there are many methodological differences between those and our meta-analysis. A recent Cochrane review [22] also investigated the administration of G-CSF in patients with liver disease. However, the study examined data dichotomously and included patients with ACLF and decompensated and compensated liver cirrhosis, without focusing on ACLF. In many of the included studies, participants also received additional pharmacological interventions that may have influenced the outcomes. Our study is the first to exclusively use data from RCTs that evaluated the effect of G-CSF in adults with ACLF, as defined by EASL-CLIF or APASL criteria, without the simultaneous administration of medications that may lead to increased plasma cell elimination, such as erythropoietin. Another unique feature of our study is its exclusive focus on patients with ACLF and the omission of studies that included acute alcoholic hepatitis. Our meta-analysis was also the first to evaluate overall survival as a hazard ratio analysis, while previous meta-analyses analyzed survival using binary data and could only comment on mortality [21,33,34]. In patients with ACLF, time plays a crucial role, as liver transplantation is the ultimate solution for many of these patients, and any other conservative treatment only serves as bridging therapy. Therefore, evaluating the therapeutic benefit while considering the time factor is of paramount importance in clinical practice. 

Specifically, our analysis demonstrated a significant benefit regarding overall survival, and patients who received G-CSF had a 38% lower risk of mortality within 60 days. When the entire observation period was taken into account, mortality was reduced by 37%. Mortality was also evaluated as a binary outcome, and a relevant advantage of patients treated with G-CSF compared to patients treated with SMT was showcased. This contrasts with some earlier meta-analyses that failed to demonstrate a benefit [33,34]. The MELD and Child–Pugh scores are validated as reliable indicators for classifying the severity of the liver disease. They are used in everyday clinical practice to predict the risk of all-cause mortality [35] and short-term survival in patients with end-stage liver disease. Our meta-analysis revealed that the administration of G-CSF therapy may have a positive impact on MELD scores, but the administration of G-CSF was not associated with an improvement in the Child–Pugh score. Both results were characterized by high heterogeneity.

All five studies reported complications associated with cirrhosis. Although our findings suggest that G-CSF therapy may reduce the incidence of complications related to cirrhosis in patients with ACLF by approximately 50% compared to those who received SMT, it is important to note that we did not observe statistically significant findings. Data on G-CSF-related adverse events were incomplete in most studies, and each study defined adverse events differently. In general, G-CSF did not lead to an increased number of infections and was well tolerated, with the exception of the study by Engelmann et al. [19], which showed the highest incidence of adverse events.

The specific mechanism by which G-CSF exerts its clinical benefit is not fully understood. ACLF is pathophysiologically characterized by two main factors: hemodynamic changes that occur within the framework of cirrhosis and involve a complex interplay between humoral changes and the fibrotic transformation of the liver parenchyma and systemic inflammatory reactions [36]. The latter includes the increased activity of monocytes, macrophages, and CD8-T cells, which is associated with elevated levels of pro- and anti-inflammatory cytokines such as TNF-α, IL-6, and IFN-γ [8,36,37]. This exaggerated immune response leads to the further activation of T-cells, which, via the release of IFN, contributes to the exacerbation of liver damage [8,36]. At the core of this immune dysregulation is a triggering event that can lead to the release of damage-associated molecular patterns (DAMPs) from parenchymal cells in all tissues, followed by the activation of the immune response [38]. Conversely, a limited function of phagocytic cells, especially neutrophils, is observed, leading to immune paralysis and the development of bacterial infections characterized by the release of toxins and virulence factors, so-called pathogen-associated molecular patterns (PAMPs) or molecular “foreign signatures”, which provide an independent pathway for the activation of defence mechanisms [39,40,41,42]. PAMPs and DAMPs and their immunological effects, in addition to vasodilation in the splanchnic region, are responsible for the development of SIRS/sepsis as clinical syndromes of a generalized inflammatory response of the body [43]. The direct and indirect liver damage caused by these mechanisms constitutes a significant part of the pathophysiology and subsequent mortality observed in patients with ACLF [40,44,45]. G-CSF can mobilize stem cells from the bone marrow, which can then migrate to the liver and differentiate into mature hepatocytes, helping with the stimulation of liver regeneration [46]. Additionally, G-CSF has been found to improve the local microenvironment of the liver, reduce liver injury and improve neutrophil activity, which is impaired in the context of ACLF and tends to result in sepsis [46]. These positive effects translate into improved liver function, decreased risk of complications, reduced risk of infection, the amelioration of histological liver damage, the support of the regeneration process, and an improvement in overall survival, as already showcased in animal models [17,18,46,47,48]. However, these theoretical benefits could not be replicated in all clinical studies, and it is important to explore the underlying cause of this. From a pathophysiological perspective, a recent study [49] investigated the hypothesis that G-CSF administration could even exacerbate ACLF by releasing bone-derived inflammatory cells that interact with Toll-like receptor 4 (TLR4) on hepatocytes, leading to an intensified inflammatory response. The results of the study showed that inhibiting TLR4 via the administration of TLR4 antagonists could reduce liver damage, improve hepatocyte proliferation, and decrease the inflammatory cascade. 

When considering the included studies, one important point stands out: Although all studies were of the very highest quality, there is a major difference in the outcome of the European study [19] compared to the Asian studies [29,30,31,32] in the sense that G-CSF showed significant efficacy in the Asian study but not in the European study. There are several possible reasons for this difference. The most obvious would be that this discrepancy is based on the different criteria for defining ACLF in the studies. While both criteria have similarities, the EASL-CLIF criteria assess six organs or systems, have stricter standards, and consider extrahepatic organ/system failure, especially renal and cerebral failure, whereas the APASL criteria are more convenient to implement and focus more on clinical changes caused by liver failure, such as ascites and hepatic encephalopathy. According to the above, the EASL-CLIF criteria encompass a wide range of disease severities, unlike the APASL criteria, which cover the early phases of ACLF with better prognosis and higher chances of recovery, as well as advanced phases characterized by multiple organ failure and high short-term mortality. For this reason, it would be reasonable to assume that this difference in criteria could be the cause of the different results. However, Engelmann et al. [19] conducted a post hoc subgroup analysis of 114 patients who met the APASL criteria for ACLF but still did not observe any improvement in survival rates via the administration of G-CSF compared to SMT. 

Apart from this mentioned and initially most obvious cause for the different outcomes, there are two other plausible reasons for this discrepancy. The first relates to the underlying cause of cirrhosis or ACLF in the patients concerned. Specifically, ACLF in the Asia-Pacific region is often associated with hepatitis B virus infection, while in Western countries, non-viral liver damage is a more common cause, with alcoholic liver damage being the main protagonist. One possible reason for the improved outcomes in the APASL studies is that a majority of patients had hepatitis B or C and received specific antiviral therapies simultaneously. In viral hepatitis, the activation of the innate immune response appears to have a limited impact on the pathogenesis of liver disease and viral clearance. In contrast, the adaptive immune response, particularly the virus-specific cytotoxic T lymphocyte response, plays a crucial role in both aspects [50]. In contrast, alcohol-induced liver injury involves a complex interplay of various mechanisms. These include the impaired function of hepatocytes, imbalanced immune responses both locally and systemically, and altered communication between parenchymal and nonparenchymal cells in the liver [51,52]. Moreover, via a TLR4-mediated activation of Kupffer cells [51], fibrinogenesis is facilitated, and this could explain the diminished effect of G-CSF on patients with alcoholic liver disease. Therefore, the variability in the underlying causes of liver cirrhosis and, as a consequence, the distinct pathophysiology of liver disease and the potentially different patient profiles further should be recognized as contributing factors to heterogeneity.

The second possible reason for the difference in outcomes is also partly related to the etiology of liver disease but focuses on bone marrow (BM) and stem cells. Advanced cirrhosis and ACLF also lead to an alteration of the stem cell compartment in the bone marrow. As recently pointed out by Engelmann et al. [53], a variety of factors observed in cirrhosis can result in a reduction in the populations of hematopoietic CD34+ cells. Bihari et al. [54] showed that CD34+ hematopoietic stem cells (HSCs) increase in the early stages of cirrhosis and progressively decline with disease severity [55]. The proinflammatory cytokine stress in cirrhosis can directly stimulate HSC proliferation and differentiation. However, chronic and excessive inflammatory cytokine signaling, especially with TNF-α and IFN-γ, can negatively affect HSCs, leading to anergy or death [56,57]. As demonstrated in the study by Anad et al. [58] in which erythropoietin and G-CSF were administered to patients with decompensated cirrhosis, intact BM is critical for the regeneration of tissue damage. In fact, patients with mild–moderate ascites and those with a healthy cellular baseline BM respond better to growth factor therapy. Altered BM may be the reason for the limited success of BM cell mobilization therapy via the administration of G-CSF observed in Engelmann et al. [19]. An appropriate cause for these changes in the BM may be the aetiology of liver disease, as the study by Engelmann et al. included a large number of patients in whom liver damage was caused by alcohol consumption, and alcohol is known to be damaging to the bone marrow [59,60]. In addition, as emphasized in a previous meta-analysis [33], 70% of the patients had cardiopulmonary failure and severe sepsis at the time of enrolment. This suggests that these patients represent a subgroup of ACLF patients with severe disease trajectories, potentially advanced sepsis or steatohepatitis, and altered bone marrow architecture and function. Ultimately, the observed differences in the results of the endpoints between the studies could suggest that there may be underlying variations in patient selection, disease aetiology, and subsequent clinical management. SMT also differed vastly between the studies, as can be seen in Table 2. 

We acknowledge that this study is subject to certain limitations, such as the limited number of studies included in the analysis and the heterogeneity observed. The included studies varied in terms of their methodology, population, and inclusion–exclusion criteria, and this heterogeneity prevents us from drawing generalized conclusions for clinical practice. We chose to carry out an available case analysis [61] when dealing with binary and survival data, and we considered missing data as missing at random. This statistical approach can produce data of high quality and usefulness, but there is always the possibility of the overestimation or underestimation of effectiveness [62]. Using continuous data, the last observation was carried forward. Although these statistical approaches are well documented and produce reliable results, they may differ significantly from actual data, and the results should be treated with caution. Furthermore, it is worth noting that the main results may be prone to statistical bias, as sensitivity analyses have not consistently confirmed them. Because of the limited number of studies eligible for inclusion and the resulting scarcity of available data, it was not possible to conduct a publication bias analysis. Perhaps the greatest disadvantage of our study is the fact that most of the data, and especially survival data, were obtained from graphs. Despite the fact that this approach is recommended by Cochrane [28] and our methods are consistent with the recommended methods, it is the case that data obtained from graphs are not always accompanied by reproducibility and robustness as they are based on graphical measurements. One important point to discuss, which underscores the necessity for future homogeneous studies, is the variability in the G-CSF regimens used across the studies, as shown in Table 2. Different dosages and administration schemata were employed, such as 5 mg/kg/day for six consecutive days in two studies [29,31], 5 mg/kg/day for five consecutive days followed by administration every 3 days until completing 12 doses in two other studies [19,30], and six consecutive doses followed by administration every other day until a total of 18 doses in one study [32]. This heterogeneity in treatment protocols makes it challenging to compare the outcomes and draw definitive conclusions regarding the optimal regimen for G-CSF administration.

## 5. Conclusions

ACLF is a complex clinical condition that has been increasingly recognized in recent years and is characterized by high mortality rates. Liver transplantation is the preferred treatment, but since it is not widely available, alternative therapeutic approaches must be established as bridging therapies. Our meta-analysis has shown that the administration of G-CSF can lead to improvements in overall survival and liver function and prognosis, as evidenced by the improvement in the MELD score. However, the heterogeneity of populations, criteria, and therapeutic protocols followed in each study prevents us from making definitive recommendations. The conduct of randomized clinical trials with similar designs, inclusion criteria, SMT, and sample size is considered critical for drawing robust conclusions. We believe that the assessment of bone marrow structure before and after treatment will provide valuable information on the outcome of therapy and may potentially allow the creation of the phenotypes of patients with acute-on-chronic hepatic failure for the purpose of tailoring and personalizing treatment. 

## Figures and Tables

**Figure 1 jcm-12-06541-f001:**
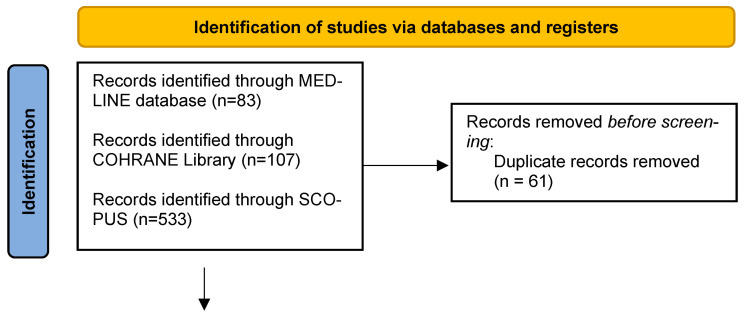
Prisma 2020 Flow diagram.

**Figure 2 jcm-12-06541-f002:**
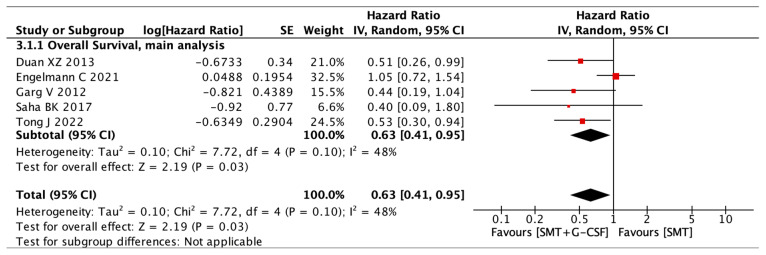
Overall survival: main analysis. Duan XZ [29], Engelmann C [19], Garg V [30], Saha BK [31], Tong J [32].

**Figure 3 jcm-12-06541-f003:**
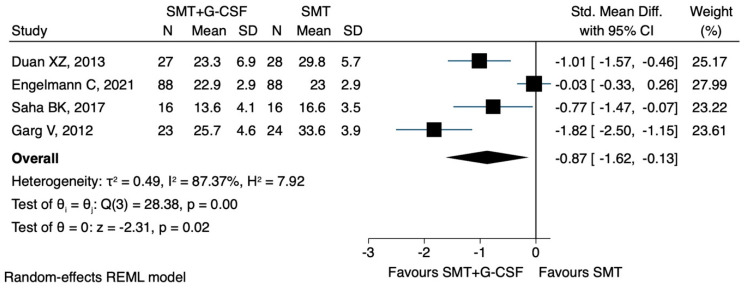
MELD score: main analysis. Duan XZ [29], Engelmann C [19], Garg V [30], Saha BK [31].

**Figure 4 jcm-12-06541-f004:**
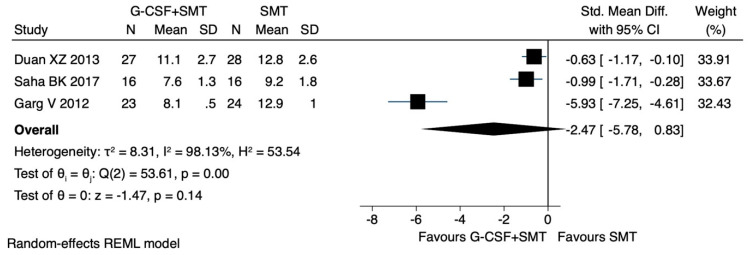
Child–Pugh score: main analysis. Duan XZ [29], Garg V [30], Saha BK [31].

**Figure 5 jcm-12-06541-f005:**
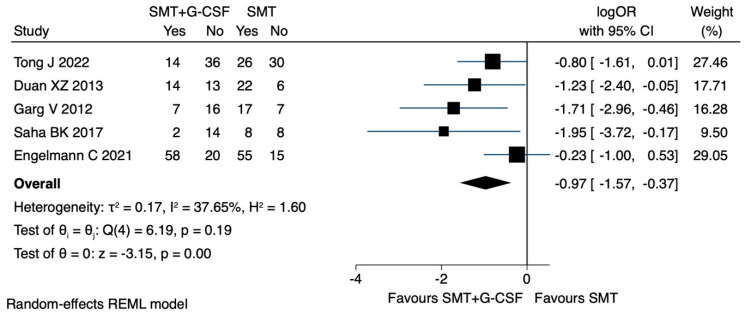
Mortality: main analysis. Duan XZ [29], Engelmann C [19], Garg V [30], Saha BK [31], Tong J [32].

**Figure 6 jcm-12-06541-f006:**
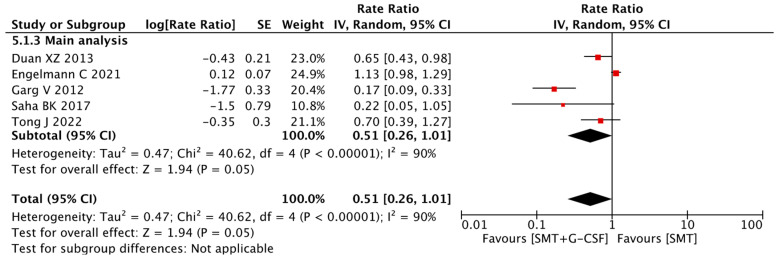
Cirrhotic complications: main analysis, Duan XZ [29], Engelmann C [19], Garg V [30], Saha BK [31], Tong J [32].

**Table 1 jcm-12-06541-t001:** Basic characteristics and summary of trials.

Author (Year), Country	Duration of Study	Number of Patients (Male %)	Age of Patients in SMT + G-CSF Group	Age of Patients in SMT Group	Cause of Cirrhosis	MELD Score in SMT+ G-CSF Group	MELD Score in SMT Group
Engelmann C [19] (2021), Germany	360 days	111 (63%)	54.4 ± 10.2 *^1^	57.1 ± 9.6 *^1^	Not reported	24.4 ± 6.3	24.5 ± 6.1
Garg V [30] (2012), India	60 days	41 (87%)	40 (30–65) *^2^	40 (19–55) *^2^	Alcohol-related cirrhosis: 62% Viral-related cirrhosis: 23% Other causes of cirrhosis: 15%	29.7 ± 4.9	30.7 ± 5.1
Saha BK [31] (2017), India	90 days	28 (87.5%)	39 (18–55) *^2^	48 (22–62) *^2^	Viral-related cirrhosis: 91% Autoimmune disease-related cirrhosis: 3% Other causes of cirrhosis: 6%	25.3 ± 3.3	26.4 ± 4.6
Duan XZ [29] (2013), China	90 days	44 (80%)	43.5 (29–63) *^2^	45.9 (22–65) *^2^	Viral-related cirrhosis: 100%	25.11 ± 3.3	26.3 ± 4.1
Tong J [32] (2022), China	180 days	91 (82%)	42.5 ± 10.2 *^1^	45.3 ± 10.6 *^1^	Viral-related cirrhosis: 100%	22.8 (20.7–26) *^2^	24.1 (21.6–27.1) *^2^

*^1^ Mean ± standard deviation; *^2^ median ± range; SMT, standard medical treatment; G-CSF, granulocyte colony-stimulating factors.

**Table 2 jcm-12-06541-t002:** Characteristics of interventions.

Author (Year), Country	G-CSF Doses	Control	Standard Medical Treatment	Administration Scheme	Administration Form	Total Doses
Saha BK [31] (2017) India	5 μg/kg	Standard medical treatment	Furosemide, spironolactone, lactulose, Rifaximin, antibiotics	Administration for 6 consecutive days	Subcutaneous injection	6
Engelmann C [19] (2021) Germany	5 μg/kg	Standard medical treatment	Lactulose, L-ornithine-L-aspartate, albumin, vasopressors, antibiotics, N-acetylcysteine, antiviral therapy *	Once a day for the first 5 days. After that, every third day until completing 12 total doses	Subcutaneous injection	12
Duan XZ [29] (2013) China	5 μg/kg	Standard medical treatment	Entecavir, albumin, glutathione, glycyrrhizin, ademetionine, polyene phosphatidylcholine, alprostadil, antiviral therapy as needed	Administration for 6 consecutive days	Subcutaneous injection	6
Garg V [30] (2012) India	5 μg/kg	Standard medical treatment	Lactulose, bowel wash, albumin, fresh frozen plasma, terlipressin, antibiotics, mechanical ventilation, vasopressors, renal replacement therapy, tenofovir pentoxifylline as needed	Once a day for the first 5 days. After that, every third day until completing 12 total doses	Subcutaneous injection	12
Tong J [32] (2022) China	5 μg/kg	Standard medical treatment	Intensive care monitoring, antiviral therapy, antibiotics albumin, terlipressin as needed	Once a day for the first 6 days. After that, every other day until day 18	Subcutaneous injection	18

* Information acquired from the protocol.

## Data Availability

Not applicable.

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
