# Peer review of "Efficacy of Granulocyte Colony-Stimulating Factor in Acute on Chronic Liver Failure: A Systematic Review and Survival Meta-Analysis"

_jcm, 2023, doi:10.3390/jcm12206541_

Round 1

Reviewer 1 Report

In this study, authors assessed the efficacy of G-CSF in patients with ACLF by meta-analysis. Although studies included in the analysis were limited, the methods were well-described and the results were comprehensive. However, some issues should be addressed.

1.         When abbreviations first appeared, their full names should be presented, especially in the Abstract.

2.         There were five randomized clinical studies were included in this systematic review. Is it possible that the small number of studies may lead to biased results?

3.         Could the differences between the two criteria of APASL and EASL affect the results in this systematic review?

4.         Some sections of the results should be described in detail. For example, Search Results.

5.         Supplementary Table 1 and 2 were missing in the manuscript.

6.         Language editing is suggested.

Minor editing of English language required

Author Response

Dear Editor Dr. Milesa Blagojevic

Dear Reviewers,

On behalf of the authors please accept our sincere gratitude for your help in sending us valuable comments and the suggested corrections, which have improved the quality of our manuscript jcm-2621201 "Efficacy of Granulocyte-colony stimulating factor therapy in patients with acute on chronic liver failure: A systematic review and survival meta-analysis."

In this updated version, we have addressed all comments and we hope that you will be satisfied with the corrections and the answers to each of them. Additionally, every correction has been made visible in the manuscript file for reading convenience using “track changes” mode. Moreover, please find below an item-by-item answer to the comments.

Hoping the revised manuscript fulfils the journal’s standards, we thank you for your courtesy.

Kind regards,

The authors

Reviewers' comments and point-by-point answers:

Reviewer #1:

1

When abbreviations first appeared, their full names should be presented, especially in the Abstract.

We would like to thank the reviewer for this comment.

This issue has now been addressed and all abbreviations when first appeared, are presented with their full names.

2

There were five randomized clinical studies were included in this systematic review. Is it possible that the small number of studies may lead to biased results?

We would like to thank the reviewer for this comment. Regarding the small number of clinical trials that met the inclusion criteria for our study and the potential biases, we discuss these issues in the 'Discussion' section

“Because of the limited number of studies eligible for inclusion and the resulting scarcity of available data, it was not possible to conduct a publication bias analysis.”

It should be noted that a small number of clinical trials can potentially be accompanied by publication bias. However, systematic reviews and meta-analyses by Cochrane have also dealt with a relatively small number of studies, and the guidelines do not recommend against conducting meta-analyses with a more limited number of studies.

Should the reviewer feel that different wording or additional information would be preferable, we would be willing to follow his/her new advice.

3

Could the differences between the two criteria of APASL and EASL affect the results in this systematic review?

We would like to thank the reviewer for this comment. The difference between the two criteria of APASL and EASL was addressed through the performance of sensitivity analysis, where the EASL study was excluded.

We also provide comments on the different results observed.

“The most obvious would be that this discrepancy is based on the different criteria for defining ACLF in the studies. While both criteria have similarities, the EASL-CLIF criteria assess six organs or systems, have stricter standards and consider extrahepatic or-gan/system failure, especially renal and cerebral failure, whereas the APASL criteria are more convenient to implement and focus more on clinical changes caused by liver failure, such as ascites and hepatic encephalopathy. According to the above, the EASL-CLIF criteria encompass a wide range of disease severities, unlike the APASL criteria, which cover early phases of ACLF with better prognosis and higher chances of recovery, as well as advanced phases characterized by multiple organ failure and high short-term mor-tality. For this reason, it would be reasonable to assume that this difference in criteria could be the cause of the different results. However, Engelmann et al.(19) conducted a post hoc subgroup analysis of 114 patients who met the APASL criteria for ACLF, but still did not observe any improvement of survival rates by administration of G-CSF compared to SMT.”

Should the reviewer feel that different wording or additional information would be preferable, we would be willing to follow his/her new advice.

4

Some sections of the results should be described in detail. For example, Search Results.

We would like to thank the reviewer for this comment. However, as the characteristics of the studies are analyzed in the 'Baseline characteristics' and 'Risk of bias in the included studies' sections, and furthermore, the study selection process has been described in the 'Data Collection and Extraction' section and documented in the flow diagram, we believe that providing a more extensive description would not contribute to our study. Should the reviewer believe that further details are required, we would be more than willing to consider a revision of the aforementioned sections.

5

 Supplementary Table 1 and 2 were missing in the manuscript.

       We would like to thank the reviewer for this comment. The tables had been mistakenly   labeled as Scheme 1 and 2. We have corrected this error.

6

Minor editing of English language required

We would like to thank the reviewer for this suggestion. Minor editing of English language has been performed.

Reviewer 2 Report

Dear Editor-in-Chief

This article reported the efficacy of granulocyte-colony stimulating factor therapy in patients with acute or chronic liver failure, which is an interesting topic today and well done. However, some concerns need to be addressed in this article.

1.           In the first citation, the abbreviation of the word should be preceded by the full name and then the abbreviation should be written (e.g., SIRS). Please correct them.

2.           What is the difference between your article and Huang's article (Ref. 21)?

3.           Please emphasize your novelty in the introduction section.

4.           What is the publication year of the articles included in this study?

5.           Please indicate the number of studies included in this study.

6.           Please indicate the exclusion criteria.

7.           Please include and discuss the following article.

“Baig, Muhammad, et al. "Efficacy of granulocyte colony stimulating factor in severe alcoholic hepatitis: a systematic review and meta-analysis." Cureus 12.9 (2020).”

8.           Please indicate the strengths of your study.

9.           Please include the author contribution section.

10.         The references are very old, please include more recent articles.

Minor editing of English language required.

Author Response

Dear Editor Dr. Milesa Blagojevic

Dear Reviewers,

On behalf of the authors please accept our sincere gratitude for your help in sending us valuable comments and the suggested corrections, which have improved the quality of our manuscript jcm-2621201 "Efficacy of Granulocyte-colony stimulating factor therapy in patients with acute on chronic liver failure: A systematic review and survival meta-analysis."

In this updated version, we have addressed all comments and we hope that you will be satisfied with the corrections and the answers to each of them. Additionally, every correction has been made visible in the manuscript file for reading convenience using “track changes” mode. Moreover, please find below an item-by-item answer to the comments.

Hoping the revised manuscript fulfils the journal’s standards, we thank you for your courtesy.

Kind regards,

The authors

Reviewers' comments and point-by-point answers:

Reviewer #2:

General comment

This article reported the efficacy of granulocyte-colony stimulating factor therapy in patients with acute or chronic liver failure, which is an interesting topic today and well done. However, some concerns need to be addressed in this article.

We would like to thank the reviewer for taking the time to perform this review. We are sure that changes made following the suggestions have further improved our manuscript and we stand at his/her disposal should the reviewer feel that more changes are needed.

1

In the first citation, the abbreviation of the word should be preceded by the full name and then the abbreviation should be written (e.g., SIRS). Please correct them.

We would like to thank the reviewer for this comment. We followed the advice, and now throughout the entire text, the full word is mentioned first, followed by an abbreviation.

2

What is the difference between your article and Huang's article

We would like to thank the reviewer for this comment. The differences between our study and the study by Huan et al. are numerous and involve not only methodological differences but also statistical ones.

Huan et al. included clinical studies in their research that examined acute alcoholic hepatitis. According to the definitions of EASL and APASL, acute alcoholic hepatitis does not necessarily imply ACLF, as ACLF is a distinct entity. The purpose of our systematic review and meta-analysis was to exclusively study the effect of G-CSF on ACLF.

Since the publication date of the above study, a large study using APASL criteria has been published, and also the only European study that examines the use of G-CSF in patients with ACLF was also published.

From a statistical perspective, it is recommended and statistically valid when studying survival to use measures that take into account the duration of follow-up, such as hazard ratio. The use of dichotomous measures like OR does not provide sufficient data for survival. Additionally, the use of OR cannot examine the potential use of G-CSF as a bridging to transplantation therapy. An extensive series of sensitive and exploratory subgroup analyses has been conducted. Adverse effects have been analyzed as rate ratios rather than odds ratios, as the same patient could potentially be affected by a complication more than once

Also, our study carefully examines and discusses the potential underlying causes of the observed differences between Europe and Asia, while also thoroughly exploring the effects and future implications of G-CSF

3

Please emphasize your novelty in the introduction section.

We would like to thank the reviewer for this comment. The novelty of our study ist now emphasized in the introduction:

“To our knowledge, this is the first state-of-the-art systematic review and meta-analysis that focuses solely on the effect of G-CSF on patients specifically with ACLF, as described in the APASL and EASL guidelines. The use of HR (Hazard ratio) as an effective measure was chosen over OR (Odds Ratio) for survival analysis due to its suitability for capturing survival-related outcomes.”

4

What is the publication year of the articles included in this study?

We would like to thank the reviewer for this comment. The information mentioned above, as well as all basic informations regarding the studies, are presented in Table 1.

5

Please indicate the number of studies included in this study.

We would like to thank the reviewer for this comment. The number of studies included in this study is indicated in “Results”:

Search Results

Five randomized clinical studies were included in our systematic review(19, 29-32).

6

Please indicate the exclusion criteria.

We would like to thank the reviewer for this comment. Exclusion criteria are mentioned in “Eligibility criteria:

Studies that investigated the use of G-CSF in combination with other active medications were excluded to maintain a focus on the specific effects of G-CSF. Studies that enrolled adults with acute liver failure not meeting the criteria for ACLF or patients with acute alcoholic steatohepatitis were also excluded. The APASL and EASL-CLIF definitions differ in their emphasis on liver failure and extra-hepatic organ failure, respectively. The definition of ACLF by EASL-CLIF is based on the CANONIC study(24) and includes three major characteristics: acute decompensation, organ failure, and a high 28-day mortality rate. On the other hand, the APASL-criteria define ACLF-CLIF as an acute hepatic insult leading to jaundice and coagulopathy, complicated within four weeks by ascites and/or encephalopathy, in patients with chronic liver disease or cirrhosis, and high 28-day mortality. ACLF was defined using either the ACLF-EASL or APASL criteria.

Should the reviewer feel that different wording or additional information would be preferable, we would be willing to follow his/her new advice.

7

Please include and discuss the following article.

“Baig, Muhammad, et al. "Efficacy of granulocyte colony stimulating factor in severe alcoholic hepatitis: a systematic review and meta-analysis." Cureus 12.9 (2020).”

We would like to thank the reviewer for this suggestion. We would happy to include and discuss the mentioned article. But in our opinion, this article focuses on acute alcoholic hepatitis, which could be a trigger for ACLF, but these two entities (ASH and ACLF) are characterized by significant pathophysiological and immunological differences. Therefore, we believe that the inclusion of this article would not contribute to our study's purpose, meaning the assessment of the effect of G-CSF on ACLF.

8

 Please indicate the strengths of your study.

We sincerely appreciate the reviewer's comment. It's worth noting that the strengths of our study have been discussed in the following sections:

This systematic review and meta-analysis represents the most recent and compre-hensive analysis of the efficacy and safety of G-CSF treatment in patients with ACLF. All studies included in the analysis were conducted with a randomized design, ensuring the reliability of the results and their usefulness in clinical practice. Three additional me-ta-analyses(21, 33, 34) have addressed this topic. Nonetheless, these studies may have limitations, such as heterogeneity in patient populations and treatment protocols, and there are many methodological differences between those and our meta-analysis. A re-cent Cochrane-review(22) also investigated the administration of G-CSF in patients with liver disease. However, the study examined the data dichotomously and included patients with ACLF and decompensated as well as compensated liver cirrhosis, without focusing on ACLF. In many of the included studies, participants also received additional phar-macological interventions that may have influenced the outcomes. Our study is the first to exclusively use data from RCTs that evaluated the effect of G-CSF in adults with ACLF as defined by EASL-CLIF or APASL criteria, without simultaneous administration of med-ications that may lead to increased plasma cell elimination, such as erythropoietin. An-other unique feature of our study is its exclusive focus on patients with ACLF and the omission of studies which included acute alcoholic hepatitis. Our meta-analysis was also the first to evaluate overall survival as a hazard ratio analysis, while previous me-ta-analyses analyzed survival using binary data and could only comment on mortality(21, 33, 34). In patients with ACLF, time plays a crucial role, as liver transplantation is the ultimate solution for many of these patients, and any other conservative treatment serves only as a bridging therapy. Therefore, evaluating the therapeutic benefit while consid-ering the time factor is of paramount importance in clinical practice.

Furthermore, the analysis of the effect of G-CSF as a Hazard Ratio has not been conducted yet, as we mention in the introduction following the recommendations of Reviewer 1:

“To our knowledge, this is the first state-of-the-art systematic review and meta-analysis that focuses solely on the effect of G-CSF on patients specifically with ACLF, as described in the APASL and EASL guidelines. The use of HR (Hazard ratio) as an effective measure was chosen over OR (Odds Ratio) for survival analysis due to its suitability for capturing survival-related outcomes.”

9

 Please include the author contribution section

We would like to thank the reviewer for this suggestion. Now a author-contribution section is added and reads as follows:

Conceptualization, C.P., J.R.A. and G.K.; methodology, C.P., G.T., E.K. and G.K.; software, G.K. and E.K.; validation, K.W, H.H.S, J.R.A. and G.K.; formal analysis, E.K. and G.K.; investigation, A.C.Z., D.E.M., W.S., C.P., G.T. and G.K.; data curation W.S., E.K. and G.K.; writing—original draft prepa-ration, A.C.Z., J.R.A., G.T., E.K. and G.K.; writing—review and editing, J.R.A., G.T., H.H.S., and G.K.; visualization, G.K. and J.R.A; supervision, C.P., G.T. and J.R.A.; project administration, J.R.A., G.K; funding acquisition, not applicable. All authors have read and agreed to the published version of the manuscript.

10

 The references are very old, please include more recent articles.

We would like to thank the reviewer for this comment.
The most references related to ACLF and the impact of G-CSF on ACLF are from 2013 onwards. The older references either represent landmarks in ACLF research or introduced relevant statistical methods that had a groundbreaking impact. Therefore, we believe that including newer references would not only be unwarranted but also not beneficial.Should the reviewer feel that other articles would be more preferable, we would be willing to follow his/her new advice.

11

Minor editing of English language required

We would like to thank the reviewer for this suggestion. Minor editing of English language has been performed.

Reviewer 3 Report

Here are some comments and suggestions for the authors

1.       The title is clear and informative, but it's quite long. Consider condensing it for brevity while retaining the essential information. For instance, "Efficacy of Granulocyte-Colony Stimulating Factor in Acute-on-Chronic Liver Failure: A Systematic Review and Meta-Analysis."

2.       The abstract follows a structured format, which is good. However, it can be even more concise. In the "Background" section, you could briefly mention the high mortality rate associated with ACLF.

3.       Introduction: The introduction and rationale is well-defined and provides a clear understanding of the study's purpose. However, it might be helpful to mention why confirming the efficacy of G-CSF is important or what clinical implications this confirmation could have.

4.       Methods: Provide a bit more detail on the search strategy. What databases were searched, and were any specific search terms or criteria used? Mention the inclusion and exclusion criteria in details for the trials.

5.       Results: The results are presented clearly with appropriate statistical data. It's important to note that the inclusion of effect sizes (HR, LogOR, SMD, rate ratio) is helpful, but it might be beneficial to explain these terms briefly for readers who are not familiar with them.

6.       Heterogeneity: Address the sources of heterogeneity. If possible, provide some insights or hypotheses about why heterogeneity exists among the included studies.

7.       Recommendations and Future Research: In the "Conclusions" section, you mention that conclusive recommendations cannot be made due to heterogeneity. It would be helpful to provide suggestions for future research directions or potential ways to address the heterogeneity issue.

8.       Prospero Registration: It's excellent that the study protocol has been registered with Prospero, indicating transparency and adherence to best research practices.

9.       Minor editing of English language required

 Minor editing of English language required

Author Response

Dear Editor Dr. Milesa Blagojevic

Dear Reviewers,

On behalf of the authors please accept our sincere gratitude for your help in sending us valuable comments and the suggested corrections, which have improved the quality of our manuscript jcm-2621201 "Efficacy of Granulocyte-colony stimulating factor therapy in patients with acute on chronic liver failure: A systematic review and survival meta-analysis."

In this updated version, we have addressed all comments and we hope that you will be satisfied with the corrections and the answers to each of them. Additionally, every correction has been made visible in the manuscript file for reading convenience using “track changes” mode. Moreover, please find below an item-by-item answer to the comments.

Hoping the revised manuscript fulfils the journal’s standards, we thank you for your courtesy.

Kind regards,

The authors

Reviewer #3:

1

The title is clear and informative, but it's quite long. Consider condensing it for brevity while retaining the essential information. For instance, "Efficacy of Granulocyte-Colony Stimulating Factor in Acute-on-Chronic Liver Failure: A Systematic Review and Meta-Analysis."

We would like to thank the reviewer for this comment. The title now reads:

“Efficacy of Granulocyte-colony stimulating factor in Acute -on-Chronic liver failure: A systematic review and survival meta-analysis”

2

The abstract follows a structured format, which is good. However, it can be even more concise. In the "Background" section, you could briefly mention the high mortality rate associated with ACLF.

We would like to thank the reviewer for this comment. The “Background” section now reads as follows:

Acute-on-chronic liver failure (ACLF) mostly occurs when there is an acute insult to the liver in patients with pre-existing liver disease, and it is characterized by a high mortality rate. Various therapeutic approaches have been used thus far, with orthotopic liver transplantation being the only definitive cure. Clinical trials and meta-analyses have investigated the use of granulocyte colony-stimulating factor (G-CSF) to mobilize bone marrow-derived stem cells. Some studies have suggested that G-CSF may have a significant role in the management and survival of patients with ACLF. However, the results are conflicting, and the efficacy of G-CSF still needs to be confirmed.

3

Introduction: The introduction and rationale is well-defined and provides a clear understanding of the study's purpose. However, it might be helpful to mention why confirming the efficacy of G-CSF is important or what clinical implications this confirmation could have.

We would like to thank the reviewer for this helpful suggestion, which has been taken into account and theINTRODUCTION reads as follows:

The examination and confirmation of the safety and clinical benefits of G-CSF in ACLF patients could offer another valuable tool in the fight against ACLF. It has the potential to serve as a 'bridging to recovery' or even 'bridging to transplantation' strategy, thereby significantly enhancing the chances of survival for ACLF patients, within the realm of non-invasive treatment options.

4

Methods: Provide a bit more detail on the search strategy. What databases were searched, and were any specific search terms or criteria used? Mention the inclusion and exclusion criteria in details for the trials.

We would like to thank the reviewer for this comment. Details about the search strategy can be found in Supplementary File 1., where all search terms and criteria are included. We would like to kindly state that Inclusion and exclusion details have already been mentioned in the eligibility criteria.

Eligibility criteria

This meta-analysis included randomized controlled trials (RCTs) that compared the efficacy of G-CSF versus placebo or any other intervention in adult patients with ACLF, and reported data for the primary outcome, i.e. overall survival. Studies that investigated the use of G-CSF in combination with other active medications were excluded to maintain a focus on the specific effects of G-CSF. Studies that enrolled adults with acute liver failure not meeting the criteria for ACLF or patients with acute alcoholic steatohepatitis were also excluded. The APASL and EASL-CLIF definitions differ in their emphasis on liver failure and extra-hepatic organ failure, respectively. The definition of ACLF by EASL-CLIF is based on the CANONIC study(24) and includes three major characteristics: acute de-compensation, organ failure, and a high 28-day mortality rate. On the other hand, the APASL-criteria define ACLF-CLIF as an acute hepatic insult leading to jaundice and coagulopathy, complicated within four weeks by ascites and/or encephalopathy, in pa-tients with chronic liver disease or cirrhosis, and high 28-day mortality. ACLF was defined using either the ACLF-EASL or APASL criteria

Should the reviewer feel that different wording or additional information would be preferable, we would be willing to follow his/her new advice.

5

Results: The results are presented clearly with appropriate statistical data. It's important to note that the inclusion of effect sizes (HR, LogOR, SMD, rate ratio) is helpful, but it might be beneficial to explain these terms briefly for readers who are not familiar with them.

.

We would like to thank the reviewer for this comment. The following text has now been added to 'statistical analysis:

“In greater detail, the effect measures selected for our analysis are as follows: The Hazard Ratio,  is utilized in survival analysis and quantifies the risk ratio between two groups taking into account the exposure time. The Logarithm of the Odds Ratio is employed to gauge associations between groups when the event under examination is dichotomous. A positive LogOR indicates a higher risk in the first group than in the second group, while a negative LogOR suggests a lower risk.  Meanwhile, the Standardized Mean Difference helps measure differences between groups in studies when dealing with continuous outcomes, taking into account standard deviations. Finally, the Rate Ratio plays a crucial role in epidemiological studies, serving as a metric for comparing event rates between two distinct groups.”

6

Heterogeneity: Address the sources of heterogeneity. If possible, provide some insights or hypotheses about why heterogeneity exists among the included studies.

We would like to thank the reviewer for bringing this up. Post-hoc and predetermined subgroup and sensitivity analyses were conducted in an attempt to explore and explain heterogeneity. Possible explanations are analyzed in the following sections in the discussion.

“When considering the studies included, one important point stands out: Although all studies were of very highest quality, there is a major difference in the outcome of the European study(19) compared to the Asian studies(29-32), in the sense that G-CSF showed significant efficacy in the Asian study but not in the European study. There are several possible reasons for this difference. . . . . . For this reason, it would be reasonable to assume that this difference in criteria could be the cause of the different results.”

Apart from this mentioned and initially most obvious cause for the different out-comes, there are two other plausible reasons for this discrepancy. The first relates to the underlying cause of the cirrhosis or ACLF in the patients concerned. Specifically, ACLF in the Asia-Pacific region is often associated with hepatitis B virus infection, while in Western countries, non-viral liver damage is a more common cause with alcoholic liver damage being the main protagonist. . . . . .. Therefore, the variability in the underlying causes of liver cirrhosis and as a consequence, the distinct pathophysiology of liver disease and potentially the different patient profiles, further should be recognized as a contributing factor to the heterogeneity.

The second possible reason for the difference in outcomes is also partly related to the etiology of the liver disease but focuses on bone marrow (BM) and stem cells. Advanced cirrhosis and ACLF also lead to an alteration of the stem cell compartment in the bone marrow. . . . . . This suggests that these patients represent a subgroup of ACLF patients with severe disease trajectories, potentially advanced sepsis or steatohepatitis and an altered bone marrow architecture and function. Ultimately, the observed differences in the results of the endpoints between the studies could suggest that there may be underlying variations in patient selection, disease aetiology, and subse-quent clinical management. SMT differed also vastly between the studies as can be seen in Table 2.

Because of the limited number of studies eligible for inclusion and the resulting scarcity of available data, it was not possible to conduct a publication bias analysis.”

Should the reviewer feel that different wording or additional information would be preferable, we would be willing to follow his/her new advice.

7

Recommendations and Future Research: In the "Conclusions" section, you mention that conclusive recommendations cannot be made due to heterogeneity. It would be helpful to provide suggestions for future research directions or potential ways to address the heterogeneity issue.

We would like to thank the reviewer for bringing this up. In fact, the conduct of homogeneous studies poses a fundamental requirement for the future.. We believe that this could be achieved with the help of the following points

The conduct of randomized clinical trials with similar design, inclusion criteria, SMT, and sample size is considered critical for drawing robust conclusions. We believe that the assessment of bone marrow structure before and after treatment will provide valuable information on outcome of therapy and may potentially allow the creation of phenotypes of patients with acute-on-chronic hepatic failure for the purpose of tailoring and person-alizing treatment.

8

Prospero Registration: It's excellent that the study protocol has been registered with Prospero, indicating transparency and adherence to best research practices.

We would like to thank the reviewer for his/her kind words and for taking the time to perform this review

9

Minor editing of English language required

We would like to thank the reviewer for this suggestion. Minor editing of English language has been performed.

Round 2

Reviewer 1 Report

All issues have been addressed.